# SARS-CoV-2 Antigen Detection to Expand Testing Capacity for COVID-19: Results from a Hospital Emergency Department Testing Site

**DOI:** 10.3390/diagnostics11071211

**Published:** 2021-07-05

**Authors:** Giulia Menchinelli, Giulia De Angelis, Margherita Cacaci, Flora Marzia Liotti, Marcello Candelli, Ivana Palucci, Rosaria Santangelo, Maurizio Sanguinetti, Giuseppe Vetrugno, Francesco Franceschi, Brunella Posteraro

**Affiliations:** 1Dipartimento di Scienze Biotecnologiche di Base, Cliniche Intensivologiche e Perioperatorie, Università Cattolica del Sacro Cuore, 00168 Rome, Italy; giulia.menchinelli@unicatt.it (G.M.); giulia.deangelis78@gmail.com (G.D.A.); margherita.c86@gmail.com (M.C.); floramarzialiotti@gmail.com (F.M.L.); ivana.palucci@unicatt.it (I.P.); rosaria.santangelo@unicatt.it (R.S.); brunella.posteraro@unicatt.it (B.P.); 2Dipartimento di Scienze di Laboratorio e Infettivologiche, Fondazione Policlinico Universitario A. Gemelli IRCCS, 00168 Rome, Italy; 3Dipartimento di Scienze dell’Emergenza, Anestesiologiche e della Rianimazione, Fondazione Policlinico Universitario A. Gemelli IRCCS, 00168 Rome, Italy; marcello.candelli@policlinicogemelli.it (M.C.); francesco.franceschi@unicatt.it (F.F.); 4Dipartimento di Sicurezza e Bioetica, Università Cattolica del Sacro Cuore, 00168 Rome, Italy; giuseppe.vetrugno@unicatt.it; 5Risk Management Unit, Fondazione Policlinico Universitario A. Gemelli IRCCS, 00168 Rome, Italy; 6Dipartimento di Medicina e Chirurgia Traslazionale, Università Cattolica del Sacro Cuore, 00168 Rome, Italy; 7Dipartimento di Scienze Mediche e Chirurgiche, Fondazione Policlinico Universitario A. Gemelli IRCCS, 00168 Rome, Italy

**Keywords:** COVID-19, emergency department, SARS-CoV-2, antigen-based testing, molecular-based testing

## Abstract

Background: SARS-CoV-2 antigen detection has currently expanded the testing capacity for COVID-19, which yet relies on the SARS-CoV-2 RNA RT-PCR amplification. Objectives: To report on a COVID-19 testing algorithm from a tertiary care hospital emergency department (ED) that combines both antigen (performed on the ED) and RT-PCR (performed outside the ED) testing. Methods: Between December 2020 and January 2021, in a priori designated, spatially separated COVID-19 or non-COVID-19 ED areas, respectively, symptomatic or asymptomatic patients received SARS-CoV-2 antigen testing on nasopharyngeal swab samples. Antigen results were promptly accessible to guide subsequent, outside performed confirmatory (RT-PCR) testing. Results: Overall, 1083 (100%) of 1083 samples in the COVID-19 area and 1815 (49.4%) of 3670 samples in the non-COVID-19 area had antigen results that required confirmation by RT-PCR. Antigen positivity rates were 12.4% (134/1083) and 3.7% (66/1815), respectively. Compared to RT-PCR testing results, sensitivity, specificity, positive predictive value (PPV), and negative predictive value (NPV) of antigen testing were, respectively, 68.0%, 98.3%, 88.8%, and 94.1% in the COVID-19 area, and 41.9%, 97.3%, 27.3%, and 98.6% in non-COVID-19 area. Practically, RT-PCR tests were avoided in 50.6% (1855/3670) of non-COVID-19 area samples (all antigen negative) from patients who, otherwise, would have needed antigen result confirmation. Conclusions: Our algorithm had value to preserve RT-PCR from avoidable usage and, importantly, to save time, which translated into a timely RT-PCR result availability in the COVID-19 area.

## 1. Introduction

In the context of a raging pandemic due to the severe acute respiratory syndrome coronavirus 2 (SARS-CoV-2), known as the etiological agent of coronavirus disease 2019 (COVID-19), COVID-19 testing strategies are crucial for the timely identification and/or management of persons with SARS-CoV-2 infection [1,2]. Since Europe has become the epicenter of the pandemic as of March 2020, the emergency department (ED) in many Italian hospitals is struggling with an overwhelming influx of persons who need daily, prompt, and accurate testing for COVID-19 [3]. This generates many samples—preferably nasopharyngeal swab samples—that daily reach hospital laboratories for COVID-19 diagnosis, which mostly relies on the amplification of SARS-CoV-2 RNA using reverse-transcription polymerase chain reaction (RT-PCR) [4].

With RT-PCR testing, results are normally available in a few hours, but test result turnaround times may be considerably delayed [5], thereby increasing the risk of SARS-CoV-2 transmission [6]. As an alternative to RT-PCR based molecular testing [4], SARS-CoV-2 antigen-based testing may limit this risk, especially in congregate settings such as a crowded ED, where confirming or excluding SARS-CoV-2 infection with reasonable timing has become essential [7,8,9]. Despite not being as sensitive as RT-PCR [4], antigen testing can be highly specific, deployed outside the clinical microbiology laboratory, and, importantly, provide a result within 15–30 min [10]. This is thanks to a lateral-flow technology that allows SARS-CoV-2 antigen to be detected and visualized as an immunoassay reactive band directly on a small portable device [11]. As of November 2020, two antigen-detection tests such as the SD Biosensor (South Korea) and the Abbott Panbio (USA) have received the WHO Emergency Use Listing, as well as the CE-marking as in vitro diagnostic medical devices [12]. A SD Biosensor antigen-detection test, namely the STANDARD F COVID-19 Ag fluorescent immunoassay (FIA), requires an instrument to readout results [2].

As in other testing sites [1,2], in EDs it is necessary to confirm antigen-positive or antigen-negative results by RT-PCR in patients who, respectively, have a low (<10%; including patients asymptomatic or symptomatic by more than 7 days after symptom onset) or high (>10%; including patients symptomatic by less than 7 days after symptom onset) probability of testing positive. Otherwise, antigen-positive results would allow patients presenting with COVID-19-compatible symptoms to wait for RT-PCR testing separated/isolated until admission to a COVID-19- or non-COVID-19-designated ward or self-isolation at home. Whether and/or how such a mitigation strategy fits the ED setting is unclear.

Here, we report on an implemented COVID-19 testing algorithm in the ED of a large tertiary care teaching hospital in Rome, Italy. Upon access, patients were triaged to two distinct ED areas, one for symptomatic patients and another for asymptomatic patients, to receive SARS-CoV-2 antigen testing using the SD Biosensor STANDARD F COVID-19 Ag FIA test. Antigen results were immediately accessible to guide confirmatory (RT-PCR) testing and/or clinical management decision-making. We compared the two areas in terms of rates of antigen results that needed RT-PCR confirmation, as well as we provided a rate estimation of RT-PCR tests avoided or adjudicated avoidable based on the implemented algorithm.

## 2. Materials and Methods

We assessed COVID-19 testing results for nasopharyngeal swab samples from patients who presented to the ED of our hospital—the Fondazione Policlinico Universitario A. Gemelli IRCCS—during a two-month period (from December 2020 to January 2021). In this period, which approximately overlaps the second/third waves of the COVID-19 epidemic in Italy [13], the ED has performed an average of 139.7 accesses per day, resulting into an average of 55.5 patients daily admitted to hospital wards. At the end of November 2020, we started detecting SARS-CoV-2 antigen with the above-mentioned SD Biosensor STANDARD F COVID-19 Ag FIA test directly in the ED to diagnose/screen patients for SARS-CoV-2/COVID-19 [14]. The ED medical and nursing staff had been trained on antigen testing before the ED director decided to implement antigen-based rapid diagnostics daily. To this end, an operating procedure for patient triage had been established as depicted in Figure 1. This procedure was compliant with the Lazio Italian Region document dated July 2020 (REGIONE.LAZIO.REGISTRO UFFICIALE.U.0577207.01-07-2020) and, later, updated in October 2020 (REGIONE.LAZIO.NOTA.0900007.21-10-2020), and with the guidelines issued by the Italian National Institute of Health in October 2020 (https://www.iss.it/documents/20126/0/COVID+19_+test+v4k_last.pdf/9ab1f211-7d88-bcb1-d454-cfed04aa8b05?t=1604483686312. Accessed on 12 May 2021).

Before antigen testing, patients could enter one of two distinct (spatially separated) areas, a priori designated as “COVID-19” or “non-COVID-19”, depending on whether patients had signs or symptoms (i.e., clinical illness) suggestive of COVID-19 [15] (Appendix A). After antigen testing—which was performed by a nurse at bedside, according to SD Biosensor manufacturer’s recommendations—patients underwent RT-PCR testing on subsequent nasopharyngeal swab samples if they were (i) symptomatic, (ii) asymptomatic with known or suspected exposure to SARS-CoV-2, or (iii) asymptomatic without known or suspected exposure to SARS-CoV-2 for whom RT-PCR was required based on clinical discretion. This included cases in which RT-PCR results affected patient isolation/quarantine decisions or dictated patients’ eligibility for surgical or medical interventions requiring hospitalization (Figure 1). The type of SARS-CoV-2 exposure was categorized as high or low risk as defined elsewhere [16]. For subsequent RT-PCR testing, samples were collected within an average of ~10 h from the initial (antigen) testing and with a 1 to 2 h range in the COVID-19 area. Samples were then sent to the clinical microbiology laboratory of the same hospital, where SARS-CoV-2 RNA detection was performed using the Seegene Allplex™ 2019-nCoV, the DiaSorin Simplexa™ COVID-19 Direct or the Roche Diagnostics Cobas^®^ SARS-CoV-2 Test [17] or the Hologic Aptima SARS-CoV-2 [18] assays. RT-PCR results were actionable to ED physicians within a mean (± standard deviation (SD)) time of ~4 ± 2 h. We compared the SD Biosensor STANDARD F COVID-19 Ag FIA test results with those of the RT-PCR assay (which was used as the reference method) to calculate performance parameters.

## 3. Results

We documented the SD Biosensor STANDARD F COVID-19 Ag FIA use for SARS-CoV-2 antigen testing on 4753 nasopharyngeal swab samples from ED patients during the study period (Figure 1 and Figure 2).

According to sample distribution in COVID-19 (*n* = 1083) and non-COVID-19 (*n* = 3670) areas, 1083 (100%) and 1815 (49.4%) samples, respectively, had (positive or negative) antigen results that required confirmation by RT-PCR (Figure 2). In the non-COVID-19 area, 1855 samples had (negative) antigen results that did not require confirmation by RT-PCR and were excluded from the analysis. The rate of antigen positivity was 12.4% (134/1083) in the COVID-19 area and 3.7% (66/1815) in the non-COVID-19 area, suggesting a differing prevalence of SARS-CoV-2 infection between the two areas. In summary, samples with a negative antigen result in both COVID-19 (*n* = 949) and non-COVID-19 (*n* = 1749) areas needed confirmatory RT-PCR testing (Figure 2). The mean (± SD) time to RT-PCR results was ~4 ± 2 h, implying that patients waited on average ~6 h in the isolation compartment/room of the COVID-19 area until the test results were available.

Table 1 shows antigen testing results compared to RT-PCR testing results.

In the COVID-19 area, antigen testing had 68.0% sensitivity and 98.3% specificity, yielding a positive predictive value (PPV) and negative predictive value (NPV) of 88.8% and 94.1%, respectively. In the non-COVID-19 area, antigen testing had 41.9% sensitivity and 97.3% specificity, yielding PPV and NPV of 27.3% and 98.6%, respectively. Antigen results were interpreted according to the manufacturer-established positivity threshold value of 1.00. As detailed in Appendix A, antigen detection cut-off index (COI) values ranged from 1.01 to 100.46 for true positive samples (*n* = 119) or from 0.01 to 0.94 for false negative samples (*n* = 56) in the COVID-19 area. COI values ranged from 1.36 to 118.30 for true positive samples (*n* = 18) or from 0.02 to 0.36 for false negative samples (*n* = 25) in the non-COVID-19 area. For false positive samples (15 in the COVID-19 area and 48 in the non-COVID-19 area), COI values ranged from 1.06 to 10.45 and from 1.01 to 22.46, respectively.

Using the above-mentioned sensitivity and specificity values, we explored the relationship between predictive values (PPV and NPV) and the probability of testing positive (i.e., the disease prevalence) in both COVID-19 and non-COVID-19 areas (Appendix A). At low pre-test probabilities of disease (prevalence rate, 0.5 to 2), PPVs and NPVs were extremely low (17.1% and 7.21%) and extremely high (99.8% and 99.7%), respectively. As probabilities increased (prevalence rate, 10 to 25), PPVs increased to 93.2% and 83.7%, respectively, and NPVs decreased to 90.2% and 83.4%, respectively. At indicated prevalence rates (Appendix A), values mirrored those observed in the COVID-19 area (pre-test probability, 16.2%) and in the non-COVID-19 area (pre-test probability, 2.4%), respectively.

As depicted in Figure 1, all patients in the COVID-19 area (*n* = 1083) as well as patients with an antigen positive result in the non-COVID-19 area (*n* = 66) underwent RT-PCR testing before being definitely adjudicated as COVID-19 positive (*n* = 193) or negative (*n* = 956) cases. Conversely, patients with a negative antigen result in the non-COVID-19 area (*n* = 3604) were adjudicated as recipients or non-recipients of RT-PCR testing if they had high-risk or low-risk exposure to a confirmed/suspected case of COVID-19, respectively. Among high-risk patients, 415 patients were definitely adjudicated as COVID-19 positive (*n* = 24) or negative (*n* = 391) cases by RT-PCR testing. Among low-risk patients, 1334 of 3189 patients were adjudicated as recipients of RT-PCR testing as hospitalization was required, leading to COVID-19 positive (*n* = 1) or negative (*n* = 1333) cases.

Theoretically, 1.4 (0.07%) of 1855 non-COVID-19 area patients would have tested positive with RT-PCR, if RT-PCR testing were used instead of antigen testing. In the COVID-19 area, 134 (12.4%) of 1083 patients would have not received RT-PCR testing if this were used to complement antigen testing (i.e., only performed on antigen negative samples). Implementing our COVID-19 testing algorithm allowed us to avoid RT-PCR tests for 1855 samples, which account for 64.0% (1855/2898) of samples tested in the ED and 7.0% (1855/26351) of samples tested in both ED (*n* = 2898) and non-ED (*n* = 23,453) hospital wards (Appendix A).

## 4. Discussion

Taking advantage of current antigen-based testing strategies to cope with COVID-19 [19], we identified two areas in the ED that differed with respect to the pre-test probability (high or low, respectively) of patient samples (*n* = 4753, in total) being tested with the SD Biosensor STANDARD F COVID-19 Ag FIA. The probability relied on the presence or absence of symptoms consistent with COVID-19, respectively, which defined the clinical context of the antigen test recipients. We found that the SD Biosensor STANDARD F COVID-19 Ag FIA displayed excellent PPV and NPV, respectively, in the COVID-19 area (88.8%, versus 27.3% in the non-COVID-19 area) and the non-COVID-19 area (98.6%, versus 94.1% in the COVID-19 area) compared to RT-PCR. This implied that, at the SD Biosensor STANDARD F COVID-19 Ag FIA testing, 15 samples in the COVID-19 area and 48 samples in the non-COVID-19 area were falsely antigen positive whereas 56 samples in the COVID-19 area and 25 samples in the non-COVID-19 area were falsely antigen negative. Despite mirroring previously reported values with the SD Biosensor antigen assay [7,8,9], a sensitivity of 68.0% in the COVID-19 area or of 41.9% in the non-COVID-19 area would imply antigen-negative results with respect to the clinical context. This is especially important among asymptomatic patients, for whom complementing an antigen-based screening strategy with RT-PCR testing could miss SARS-CoV-2 infected patients [20].

If the assumed COVID-19 prevalence is 20% or less, as in our study, the NPV of an antigen test (like the SD Biosensor STANDARD F COVID-19 Ag FIA) is such that a negative result may allow us to exclude the disease with >97.5% confidence [21]. However, lowest false-negative result rates occur in contexts of low disease prevalence and the high test’s sensitivity that is not exactly the case of antigen tests currently available [10]. Meanwhile, improving the sensitivity of these tests specifically [21] or, rather, the “sensitivity of the testing regimen” [22] would reduce the risk that persons with false-negative antigen results spread the disease. We mitigated the unintended consequences of our testing strategy by keeping patients in the COVID-19 area isolated from one another until subsequent RT-PCR testing provided confirmation of SD Biosensor STANDARD F COVID-19 Ag FIA results. Indeed, isolation of patients in the COVID-19 area compartments/rooms was in spite of whether they had positive or negative antigen results. Likewise, patients in the non-COVID-19 area who tested positive with the SD Biosensor STANDARD F COVID-19 Ag FIA were pre-emptively moved close to the COVID-19 area (and, consequently, isolated) unless they had a low risk of SARS-CoV-2 exposure and a prospect of being isolated as in the COVID-19 area (Appendix A). Studies have shown that ED—the hospital gatekeeper—may become the “epicenter” of hospital-associated outbreaks due to droplet- or contact-transmitted respiratory viruses [23]. Unsurprisingly, intensified access of patients and ensuing turmoil (especially in daytime working hours) in the ED steadily menace the effectiveness of strategies to screen, isolate, and test for suspected COVID-19 [24].

Besides improving the management of patients in the ED, our testing strategy led to decreased numbers of RT-PCR tests in those patients who, otherwise, would have needed confirmation of previous antigen results. Consistent with previous findings [7], we assumed that asymptomatic persons triaged to the non-COVID-19 area were most likely not infectious. Indisputably, RT-PCR is the benchmark test for detecting SARS-CoV-2 infection [22]. However, lateral-flow antigen tests, similar to the SD Biosensor STANDARD F COVID-19 Ag FIA [4], are capable to identify infected persons in whom SARS-CoV-2 has grown enough to become detectable and, consequently, who are most likely infectious [21]. We experienced that meeting the benchmark analytical limit of detection—that is 100 or 1000 times higher for antigen tests [22]—may be unnecessary. In our previous study [25], the sensitivity of the SD Biosensor STANDARD F COVID-19 Ag FIA was sufficient only with nasopharyngeal swab samples with RT-PCR cycle threshold (Ct) values lower than 25, i.e., corresponding to higher SARS-CoV-2 RNA levels, which are likely to occur within the first seven days of SARS-CoV-2 infection. Therefore, it is plausible that only few (1 to 2) of 1855 patients, for whom ED physicians did not require RT-PCR testing for confirmation, would subsequently test RT-PCR positive for SARS-CoV-2. It is also plausible that detecting SARS-CoV-2 RNA in those patients signified to identify persons who had passed the SARS-CoV-2 infectious (then transmissible) period who had still high viral loads (≥10^6^ copies/mL) and then were prone to transmit SARS-CoV-2, or who, as recently observed [26], had low viral loads (<10^5^ copies/mL) but were able to originate clusters of SARS-CoV-2 infection. Nonetheless, avoiding nearly 70% of RT-PCR tests in the non-COVID-19 area had value not only in terms of cost saving, but also of time saving, thereby translating into faster notified RT-PCR results by the laboratory staff to the ED physicians in the COVID-19 area.

Our study is a retrospective assessment of a COVID-19 testing algorithm that used (albeit at a different extent) both antigen—the SD Biosensor STANDARD F COVID-19 Ag FIA—and RT-PCR tests. Unlike [27], or like [7,8,9], we focused on the ED due to its suitability as a clinical context to “test the sensitivity” of a COVID-19 testing strategy. As already observed [7], comprising symptomatic and asymptomatic patients, ED settings display a COVID-19 prevalence similar to that in the general population (<10%). Besides assessing SD Biosensor STANDARD F COVID-19 Ag FIA performance, we estimated the algorithm as a whole, calculating the number of avoided RT-PCR tests upon its implementation over the study period. Accordingly, we did not analyze the impact of this implementation in terms of accidental SARS-CoV-2 exposure rates for ED workers, or nosocomial COVID-19 transmission in our hospital. However, we are confident that these events do not occur so frequently [24]. Since the second wave of COVID-19 in Italy—which our study refers to—the ED in our hospital undertook deep reorganization [3], which fundamentally consisted into separating patient entries before accessing to dedicated COVID-19 or non-COVID-19 areas. To date, transmission-related clusters of COVID-19 have been identified in our hospital but, notably, none of these clusters was related to previous long stays (≥72 h) of patients in the ED. A limitation of our study is that we did not correlate antigen results (only in symptomatic patients) with the days after symptom onset as well as antigen results with RT-PCR Ct values. Although this information might be crucial to weigh the risks and consequences of false positive and false negative results obtained with the SD Biosensor STANDARD F COVID-19 Ag FIA, the study design hampered a complete data retrieval for all the samples included in the analysis. Thus, stratifying antigen testing results by RT-PCR Ct values/viral loads in this study might have allowed better appreciation of the SD Biosensor STANDARD F COVID-19 Ag FIA performance and added further support to previously published data on this topic.

## 5. Conclusions

Rapid diagnosis of SARS-CoV-2 infection remains essential for our daily battle against COVID-19. Until more data is accumulating on alternative diagnostic tests, RT-PCR based molecular testing will be largely used in the next months in many hospitals. However, in contexts such as the ED here described, use of antigen-based testing (i.e., followed or not followed by confirmatory RT-PCR) may be advantageous. Our findings support the growing evidence on the importance of incorporating rapid antigen tests, including the SD Biosensor STANDARD F COVID-19 Ag FIA, in COVID-19 testing strategies. In ED settings, where “time is of essence”, antigen testing may therefore allow timely management of patients who need COVID-19 diagnosis and, meanwhile, preservation of RT-PCR from unnecessary (and time-expensive) usage.

## Figures and Tables

**Figure 1 diagnostics-11-01211-f001:**
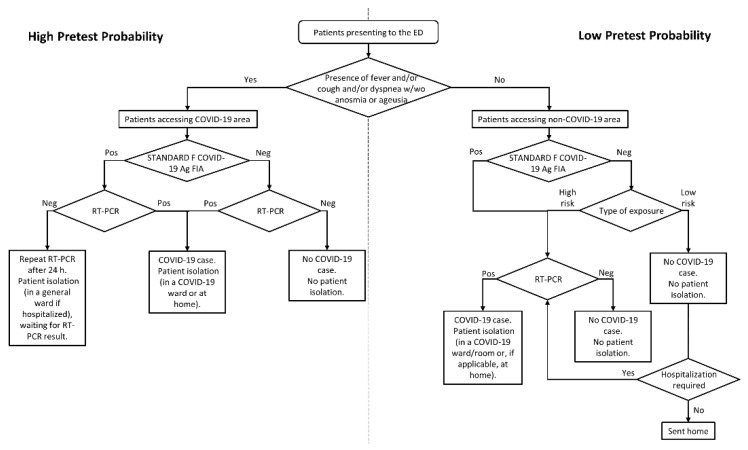
Testing algorithm for diagnosing SARS-COV-2 infection in patients who presented to the ED from December 2020 to January 2021. Depending on whether the patients were symptomatic or asymptomatic, two areas were designated as COVID-19 or non-COVID-19 to represent high or low pre-test probability contexts, respectively. In the non-COVID-19 area, patients were subdivided in high risk or low risk categories according to the type of exposure to SARS-CoV-2 (as defined in the text). The SD Biosensor STANDARD F COVID-19 Ag FIA (performed on the ED) or the RT-PCR (performed outside the ED) tests allowed detection of SARS-COV-2 antigen or RNA, respectively, in patients’ nasopharyngeal swab samples. Abbreviations: ED, emergency department; FIA, fluorescent immunoassay; COVID-19, coronavirus disease 2019; SARS-CoV-2, severe acute respiratory syndrome coronavirus 2; and RT-PCR, reverse-transcription polymerase chain reaction.

**Figure 2 diagnostics-11-01211-f002:**
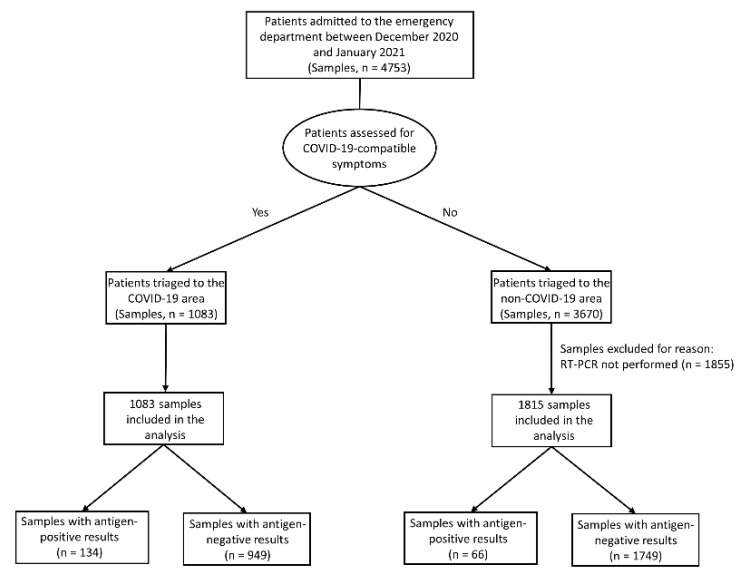
Flowchart of nasopharyngeal swab samples from patients triaged to COVID-19 or non-COVID-19 areas of the ED. Samples were tested for SARS-CoV-2 antigen with the SD Biosensor STANDARD F COVID-19 Ag FIA, yielding positive or negative results. Antigen results were compared with those obtained by RT-PCR for cases (1083 in the COVID-19 area, and 1815 in the non-COVID-19 area) in which confirmatory RT-PCR testing (performed within few hours from antigen testing) was required. Abbreviations: ED, emergency department; FIA, fluorescent immunoassay; COVID-19, coronavirus disease 2019; SARS-CoV-2, severe acute respiratory syndrome coronavirus 2; and RT-PCR, reverse-transcription polymerase chain reaction.

**Table 1 diagnostics-11-01211-t001:** Antigen test performance according to the RT-PCR positivity or negativity used to determine the positive or negative COVID-19 status ^a^.

	Antigen Test by COVID-19 Status (Positive or Negative)			
	COVID-19 Area		Non-COVID-19 Area	
Parameter	Positive	Negative	Positive	Negative
Positive results ^b^	119	15	18	48
Negative results ^c^	56	893	25	1724
Sensitivity, % (95% CI)	68.0 (60.5–74.8)	…	41.9 (27.0–57.9)	…
Specificity, % (95% CI)	98.3 (97.3–99.1)	…	97.3 (96.4–98.0)	…
Predictive value, % (95% CI)				
Positive	88.8 (82.2–93.6)	…	27.3 (17.0–39.6)	…
Negative	94.1 (92.4–95.5)	…	98.6 (97.9–99.1)	…

Abbreviations: CI, confidence interval; COVID-19, coronavirus disease 2019; and RT-PCR, reverse-transcription polymerase chain reaction. ^a^ The SD Biosensor STANDARD F COVID-19 Ag FIA test was used to detect SARS-CoV-2 antigen in the nasopharyngeal swab samples of patients who accessed COVID-19 or non-COVID-19 areas of the emergency department (see text for details). The positive or negative COVID-19 status was determined by the RT-PCR assay performed on a nasopharyngeal swab that was subsequently sampled from the patient according to clinical discretion (see text for details). ^b^ Data represent true- and false-positive results, as assessed in comparison with those obtained by the RT-PCR assay, respectively. ^c^ Data represent false- and true-negative results, as assessed in comparison with those obtained by the RT-PCR assay, respectively.

## Data Availability

The data presented in this study are available on request from the corresponding author.

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
