# Peer review of "SARS-CoV-2 Antigen Detection to Expand Testing Capacity for COVID-19: Results from a Hospital Emergency Department Testing Site"

_diagnostics, 2021, doi:10.3390/diagnostics11071211_

Round 1
Reviewer 1 Report
Dear Authors of the manuscript
Although your manuscript looked at an important problem these days it needs major revision to be published.
The main points are:
1. You did not include Ct values in your text. Ct values would clarify bad sensitivity and moderate specificity for hospital patients and even more clarify extremely bad data regarding asymptomatic patients.
Such data without Ct values will only confuse readers without the proper presentation.
2. It is not clear what the main objective of your article is?
Is that diagnostic accuracy of the test or cost-benefit analyses of using antigen test as a screening tool to reduce number of PCRs?
Please clarify.
3. Cut off in ranges does not tell us anything about the test performances. If you want to calculate true cut-off of your test you should calculate it and present it with 95% CI.
4. You used your tests outside of Instructions of use, more than 7 days since the start of the symptoms. Why did you decide to use it in such way and why this patrticulate test? Did you test any other tests in pilot study?
Regards
Author Response
Dear Authors of the manuscript
Although your manuscript looked at an important problem these days, it needs major revision to be published.
The main points are:
- You did not include Ct values in your text. Ct values would clarify bad sensitivity and moderate specificity for hospital patients and even more clarify extremely bad data regarding asymptomatic patients. Such data without Ct values will only confuse readers without the proper presentation.
Answer: As specified below (see the answer to comment 2), we did not include Ct values in the text. This information (albeit important “to weigh the risks and consequences of false positive and false negative results obtained with the SD Biosensor STANDARD F COVID-19 Ag FIA”, especially in asymptomatic patients) was necessarily omitted (see the Limitations section of the manuscript) but, fortunately, was not crucial to the main aim of our study. Indeed, the aim was not primarily to assess the performance of the SD Biosensor STANDARD F COVID-19 Ag—which has been thoroughly investigated in previous studies, including ours (see ref. 25 [formerly ref. 24])—but to show how the antigen testing allowed reducing the number of RT-PCR tests in a clinical setting such as the emergency department. We clarified this issue in the last sentence of the Introduction, and added a further comment in the Limitations section of the manuscript. See page 2, lines 85 to 88, and page 8, lines20 to 23, of the revised manuscript.
- It is not clear what the main objective of your article is?
Is that diagnostic accuracy of the test or cost-benefit analyses of using antigen test as a screening tool to reduce number of PCRs?
Please clarify.
Answer: We apologize for the unclearness of our statement about the main objective of the study. As specified above (see the answer to comment 1), the main aim of our study was to provide the evidence that using antigen testing as a “screening tool” allowed to reduce considerably the number of RT-PCR tests and, thus, to preserve RT-PCR from avoidable usage. We clarified this issue in the last sentence of the Introduction. See page 2, lines 85 to 88, of the revised manuscript.
- Cut off in ranges does not tell us anything about the test performances. If you want to calculate true cut-off of your test, you should calculate it and present it with 95% CI.
Answer: We agree with reviewer 1 that cutoff ranges do not inform about the antigen test performance. Thus, we omitted this information from Table 1 and we deleted some sentences in the text, leaving only data to comment Supplementary Figure S2. See pages 5 and 6 of the revised manuscript.
- You used your tests outside of Instructions of use, more than 7 days since the start of the symptoms. Why did you decide to use it in such way and why this particular test? Did you test any other tests in pilot study?
Answer: We decided to use the SD Biosensor STANDARD F COVID-19 Ag FIA based on our previous experience with this antigen test (see ref. 25 [formerly ref. 24]), which allowed us to show that the performance of the test vary according to lower or higher Ct values/viral loads. Consistent with our findings, the INCREASE study performed in the emergency ward by Caruana et al. (see ref. 9) showed that, regardless of the type of test used, antigen testing was less sensitive than RT-PCR testing, and this was particularly evident in either late presenters (symptom onset delay >7 days) or asymptomatic patients admitted for COVID-19. Unfortunately, in the present study, data on the symptom duration to ED admission in the patients labeled as COVID-19 symptomatic were missing (see the Limitations section of the manuscript), and this hampered us to stratify the antigen test sensitivity according to the delay since symptom onset.
Regards
Answer: We are very grateful to reviewer 1 for having appreciated our efforts in this study.
Reviewer 2 Report
To the Authors:
This is a well-designed and interesting study, corroborating previous results on the possibility of carefully incorporating rapid antigen tests to the diagnostic algorithm for patients’ triaging in the emergency department. I have few observations:
- Abstract (Lines 36-38): “Theoretically, consistent with an enough high NPV of the antigen test, very few samples from patients for whom RT-PCR was avoided would test 37 positive for SARS-CoV-2 RNA”. I would remove this sentence from the abstract, being misleading for readers without direct hands-on the AG testing and, in general, if read without supporting explanation.
- Introduction: In Line 29, “1815 (49.4%) 3670 samples” should become 1815 (49.4%) of 3670 samples.
- Introduction: In Lines 70-71, the Authors stated that “it is necessary to confirm antigen-positive results by RT-PCR in patients who have a low probability of testing positive”, but they do not discuss the need to confirm antigen-negative results in patients with a high probability of testing positive.
- Introduction: In Line 101 “Istituto Superiore di Sanità” might be translated into National Office of Public Health, in order to make it understandable to international readers.
- Methods (Line 107): how were the (antigen or PCR) test performed? Were the AG tests performed directly in the emergency department at the bedside or rather transported (in a transport medium) to the laboratory? If performed at bedside, were they performed by a nurse/laboratory technician/ physician? I believe that adding this information might increase the completeness and transparency of the procedure.
- Results: While I particularly appreciated the creation of COVID and non-COVID areas and subsequent analyses, I think that the assessment of the antigen detection cut-off index (COI) values might be misleading to the readers when comparing this study to results from similar works. For this reason, I would suggest either to better explain the added value of COI values or to limit the description of COI results and better focus on sensitivity/specificity/NPV/PPV compared to other studies.
- Discussion: Sensitivity and specificity data are showed but not adequately discussed. Overall sensitivity of 68% in COVID-19 area is sub-optimal, overall sensitivity of 41.9% in non COVID19 area is very low. While these results corroborate some previously published studies, the Authors should better address the possible problems coming from low sensitivity, especially among asymptomatic patients (Caruana et al. https://doi.org/10.1016/j.nmni.2021.100899), trying to raise awareness among readers regarding the importance of the clinical context (e.g. immunocompromised patients should always be tested with RT-PCR methods).
- Discussion (Lines 267-270): “detecting SARS-CoV-2 RNA in those patients signified to identify persons who had passed the SARS-CoV-2 infectious”: I suggest toning down this sentence, being pure conjecture, referring to it as a hypothesis. Indeed, COVID-19 asymptomatic patients might still have high viral loads, thus still transmit the virus. Furthermore, recent data demonstrated the possible onset of clusters of infections originating from patients with VLs below 105 copies/mL (Ladoy et al. PMID: 34000545), showing the risks of missed diagnoses at lower viral loads. Finally, there is no mention, in this study, of different sensitivities obtained when considering different viral loads, which, if implemented, might help to clarify this point and support previously published data on AG test performances.
Author Response
This is a well-designed and interesting study, corroborating previous results on the possibility of carefully incorporating rapid antigen tests to the diagnostic algorithm for patients’ triaging in the emergency department.
Answer: We are very grateful to reviewer 2 for having appreciated our efforts in this study.
I have few observations:
- Abstract (Lines 36-38): “Theoretically, consistent with an enough high NPV of the antigen test, very few samples from patients for whom RT-PCR was avoided would test positive for SARS-CoV-2 RNA”. I would remove this sentence from the abstract, being misleading for readers without direct hands-on the AG testing and, in general, if read without supporting explanation.
Answer: We agreed with the suggestion of reviewer 2 and we removed the sentence accordingly. See page 1 of the revised manuscript.
- Introduction: In Line 29, “1815 (49.4%) 3670 samples” should become “1815 (49.4%) of 3670 samples.”
Answer: We modified the sentence by adding “of”. See page 1 of the revised manuscript.
- Introduction: In Lines 70-71, the Authors stated, “it is necessary to confirm antigen-positive results by RT-PCR in patients who have a low probability of testing positive”, but they do not discuss the need to confirm antigen-negative results in patients with a high probability of testing positive.
Answer: As suggested, we completed the sentence by commenting on the need to confirm antigen-negative results in patients with a high probability of testing positive. See page 2, lines 70 to 75, of the revised manuscript.
- Introduction: In Line 101, “Istituto Superiore di Sanità” might be translated into National Office of Public Health, in order to make it understandable to international readers.
Answer: As suggested, we translated “Istituto Superiore di Sanità” into “National Institute of Public Health”. See page 3, lines 105 to 106, of the revised manuscript.
- Methods (Line 107): how were the (antigen or PCR) test performed? Were the AG tests performed directly in the emergency department at the bedside or rather transported (in a transport medium) to the laboratory? If performed at bedside, were they performed by a nurse/laboratory technician/ physician? I believe that adding this information might increase the completeness and transparency of the procedure.
Answer: As required, the relevant information was added. See page 3, line 113, of the revised manuscript.
- Results: While I particularly appreciated the creation of COVID and non-COVID areas and subsequent analyses, I think that the assessment of the antigen detection cut-off index (COI) values might be misleading to the readers when comparing this study to results from similar works. For this reason, I would suggest either to better explain the added value of COI values or to limit the description of COI results and better focus on sensitivity/specificity/NPV/PPV compared to other studies.
Answer: We chose to limit the description of COI values, and this choice was also in accordance with the reviewer 1’s suggestion. Consistently, we deleted the COI value information from Table 1 and some relative sentences from the text. See page 5 of the revised manuscript.
- Discussion: Sensitivity and specificity data are showed but not adequately discussed. Overall sensitivity of 68% in COVID-19 area is suboptimal; overall sensitivity of 41.9% in non-COVID19 area is very low. While these results corroborate some previously published studies, the Authors should better address the possible problems coming from low sensitivity, especially among asymptomatic patients (Caruana et al. https://doi.org/10.1016/j.nmni.2021.100899), trying to raise awareness among readers regarding the importance of the clinical context (e.g. immunocompromised patients should always be tested with RT-PCR methods).
Answer: Regarding the relevant issue highlighted by reviewer 2, we added some sentences to comment on the possible problems arisen from the low sensitivity of antigen testing, particularly in asymptomatic patients. The suggested reference was, obviously, included. See page 6, lines 46 to 51, of the revised manuscript and the modified reference list.
- Discussion (Lines 267-270): “detecting SARS-CoV-2 RNA in those patients signified to identify persons who had passed the SARS-CoV-2 infectious”: I suggest toning down this sentence, being pure conjecture, referring to it as a hypothesis. Indeed, COVID-19 asymptomatic patients might still have high viral loads, thus still transmit the virus. Furthermore, recent data demonstrated the possible onset of clusters of infections originating from patients with VLs below 105 copies/mL (Ladoy et al. PMID: 34000545), showing the risks of missed diagnoses at lower viral loads. Finally, there is no mention, in this study, of different sensitivities obtained when considering different viral loads, which, if implemented, might help to clarify this point and support previously published data on AG test performances.
Answer: As suggested, we rephrased the sentence by including all the possible scenarios with asymptomatic patients. The suggested reference was, obviously, included. Finally, in the Limitations section, we commented on the need of stratifying antigen testing results by RT-PCR Ct values/viral loads for better appreciating the SD Biosensor STANDARD F COVID-19 Ag FIA performance and adding further support to previously published data on this topic. See page 7, lines 91 to 94, and page 8, lines 20 to 23, of the revised manuscript.